# Multi-cancer analysis of histopathologic MSI screening based on digital histology image

**Jin-Ok Lee**[1]☯, **Chang Yeon Kim**[2]☯, **Sejoon Lee**[3,4,5]*, **Jin-Haeng Chung**[3,6]*

**1** Department of Health Science and Technology, Graduate School of Convergence Science and Technology, Seoul National University, Seoul, Korea, **2** Seoul National University College of Medicine, Seoul, Korea, **3** Department of Pathology and Translational Medicine, Seoul National University Bundang Hospital, Seongnam, Korea, **4** Precision Medicine Center, Seoul National University Bundang Hospital, Seongnam, Korea, **5** Department of Genomic Medicine, Seoul National University Bundang Hospital, Seongnam, Korea, **6** Department of Pathology, Seoul National University College of Medicine, Seoul, Korea

☯ These authors contributed equally to this work.
* sejoonlee@snubh.org (SJL); jhchung@snubh.org (JHC)

## Abstract

Microsatellite instability, a genetic indication of DNA mismatch impairment, provides promising treatment options. Our study aimed to detect the mutation with whole-slide image (WSI) and discover the most effective pre-trained deep-learning model to sort diagnostic slides between high microsatellite instability (MSI-H) and microsatellite stable (MSS). WSI data retrieved from public dataset were processed for training and evaluating MSI categorization model. We detected MSI in slide levels for colorectal cancer (CRC), stomach adenocarcinoma (STAD), uterine corpus, and endometrial adenocarcinoma (UCEC). Models trained with a single tissue type were evaluated with the test dataset of corresponding tissue and subsequently with the test dataset of other types of tissue (cross-tissue evaluation). Finally, another model trained with multi-tissue types was built to predict the test dataset of individual tissue. Our models achieved AUC values of 0.93, 0.84, and 0.79 in TCGA-CRC, TCGA-STAD and TCGA-UCEC, respectively. We observed that a model trained on a corresponding tumor tissue demonstrates higher accuracy, particularly compared to those trained on other tumor tissues. In the combined model trained on multi-tissue, we observed diverse outcomes regarding which model was prioritized depending on the cancer type. These results demonstrate that models trained on multiple tissues have the potential to discern features that are generalizable across different types of cancer.

## Introduction

Microsatellite instability (MSI) is a genetic presentation of hypermutability that originates from impairment of various mismatch repair (MMR) genes, such as *MLH1, MSH2, MSH6,* and *PMS2*. Dysfunction of MMR genes disrupt repairment of single

**Data availability statement:** The results published here are based on data generated by The Cancer Genome Atlas and obtained from the Database of Genotypes and Phenotypes (dbGaP) with accession number phs000178/GRU. Information about TCGA can be found at https://portal.gdc.cancer.gov/. All other remaining data are available within the article and supporting files, or available from the authors upon request.

**Funding:** This work was supported by a research fund from Seoul National University Bundang Hospital (grant no. 18-2018-0023 and 18-2025-0005). The funders had no role in study design, data collection and analysis, decision to publish, or preparation of the manuscript.

**Competing interests:** The authors have declared that no competing interests exist.

base or sequence errors triggered during replication, increasing the risk of malignant genetic changes. The MSI has been a promising genetic marker for oncologists due to its clinical significance with various tumor types [1]. Compared to other somatic mutations, the high frequency of MSI attributes it as a potential therapeutic target. PDL-1 blockade therapy, such as Pembrolizumab and nivolumab, was The US Food and Drug Administration (FDA) approved for a tumor site-agnostic solid tumor indication [2]. MSI-H has been identified in various cancer type including breast cancer, ovarian cancer and others [3]. Among various tissue types, uterine corpus endometrial carcinoma (UCEC) has the highest prevalence of high-level MSI (MSI-H) (17.00–31.37%), followed by colorectal adenocarcinoma (COAD) (6.00–19.72%), and stomach adenocarcinoma (STAD) (9.00–19.09%) [4].

However, common hindrances for diagnosing MSI are time and cost. Though MSI screening can be conducted using a Polymerase Chain Reaction (PCR) or genetic fragment analysis, a conclusive diagnosis requires the Next Generation Sequencing (NGS) or immunohistochemistry techniques [5]. However, the procedure is expensive and takes 2 weeks or more. Furthermore, the MSI incidence varies with tumor type; MSI routine screening is usually not conducted under clinical conditions [6]. Given these limitations, there is a growing need for more accessible and efficient MSI detection methods.

Over the past decade, image analysis with deep learning (DL) has drastically improved, leading to the development of numerous models that assist clinicians in formulating personalized cancer treatment strategies. These predictive models have leveraged diverse imaging modalities, including, magnetic resonance imaging (MRI) for tumor staging and treatment planning [7], and CT/PET imaging for predicting treatment response in lung cancer [8], and immunohistochemistry (IHC) images for genetic profile prediction and treatment response assessment [9]. Of particular importance, hematoxylin and eosin (H&E) stained slide images exhibit how various genetic alterations manifest as distinctive patterns in histopathological images [10]. Predictive models based on these genetic profiles enable the prediction of immunotherapy response or survival analysis. For example, MSI-H colorectal cancer histopathologic images present more tumor-infiltrating lymphocytes, mucinous differentiation, medullary-like morphology, or lack of necrosis [11,12]. Therefore, an alternative screening method using DL models to analyze MSI features from H&E slide images may provide a more rapid and cost-effective diagnosis method compared to traditional approaches. Such an approach could potentially enhance patient outcomes and improve clinical workflows.

Recent studies has been developed multiple DL models to predict MSI/dMMR status, but they have been limited to few cancer types such as STAD, UCEC, and primarily CRC [13–16]. It may be the relative availability of the training data required for building DL models compared to other cancer type. The performance of deep learning models is heavily reliant on the quality and size of the training data used. The limitation of available training data poses a significant obstacle to the development and application of deep learning models.

In this study, we aimed to develop an effective model capable of detecting MSI features across various cancer types, even when faced with limited training data. To achieve this, we plan to construct and evaluate models from diverse perspectives. First, we conduct both corresponding tissue evaluations and cross-tissue evaluations for a model trained on a each single tissue. This process will validate whether each models accurately represents the molecular features or tissue-specific characteristics of MSI. Secondly, to construct a model aimed at maximizing MSI molecular features while minimizing tissue-specific characteristics, we trained models by combining images from various tissues. This was done with the goal of improving performance through increased training data and enhancing the generalization ability across various types of cancer.

## Materials and methods

We summarized the entire method with pipeline, from data download and pre-processing to deep learning model training and evaluation (Fig 1).

### Imaging and clinical data

Using the Genomics Data Commons (GDC) Data Transfer Tool at the National Cancer Institute in Bethesda, MD, USA, we downloaded diagnostic Whole Slide Images (WSIs) from The Cancer Genome Atlas (TCGA) public database. The four TCGA projects—TCGA-COAD, TCGA-READ, TCGA-STAD, and TCGA-UCEC—were selected because they predominantly feature MSI compared to other cancer types in a multicentric collection of tissue specimens. Each label represented colon, rectal, stomach, uterine corpus, and endometrial adenocarcinoma. We combined WSIs (TCGA-CRC) for TCGA-COAD and TCGA-READ cancers due to their molecular and histological similarities [17].

The ground-truth labels of MSI in WSI were obtained from PCR test results from the GDC portal. To construct models categorizing between high microsatellite instability (MSI-H) and microsatellite stable (MSS), we utilized images with the corresponding labels. After a basic slide quality review, we selected a total of 1,282 WSIs from 1,244 patients, comprising 82, 64, and 154 patients in the MSI-H group, and 408, 260, and 276 patients in the MSS group for TCGA-CRC, TCGA-STAD, and TCGA-UCEC, respectively.

To perform external validation of our models, we utilized COAD and UCEC cohort datasets from CPTAC (Clinical Proteomic Tumor Analysis Consortium). A single WSI per patient was used for the validation, which comprised 18 MSI-H and 46 MSS patients from CPTAC-COAD, and 16 MSI-H and 58 MSS patients from CPTAC-UCEC.

### Data preprocessing

The PathProfiler package from GitHub was employed to import all svs images and convert them into 512x512-pixel PNG images (https://github.com/MaryamHaghighat/PathProfiler [18]. The pretrained Unet algorithm from PathProfiler was then utilized to filter tiles exclusively within the region of interest. However, tiles with less than 10% of the region of interest (ROI) were excluded from the analysis. The StainTools package was imported from GitHub for image normalization (github.com/Peter554/StainTools). Tiled images underwent color normalization using the Macenko method [19] to ensure matching brightness and contrast within each group, and were subsequently resized to 224 x 224 pixels to serve as the input for the DL models. During this process, tiles containing background or blurry images were automatically removed from the dataset, utilizing the detected edge quantity (Canny edge detection in Python's OpenCV package) (https://github.com/KatherLab/preProcessing).

### A tumor tile classification model development

Only tumor tiles were filtered for MSI analysis using the 'tumor or non-tumor classifying' DL model from entire tiled images. We utilized a ground-truth dataset categorized in three groups of six classes from doi.org/10.5281/zenodo.2530788: ADIMUC (adipose, muscous), STRMUS (stroma, muscle), TUMSTU (colorectal tumor, and stomach tumor) [13]. The

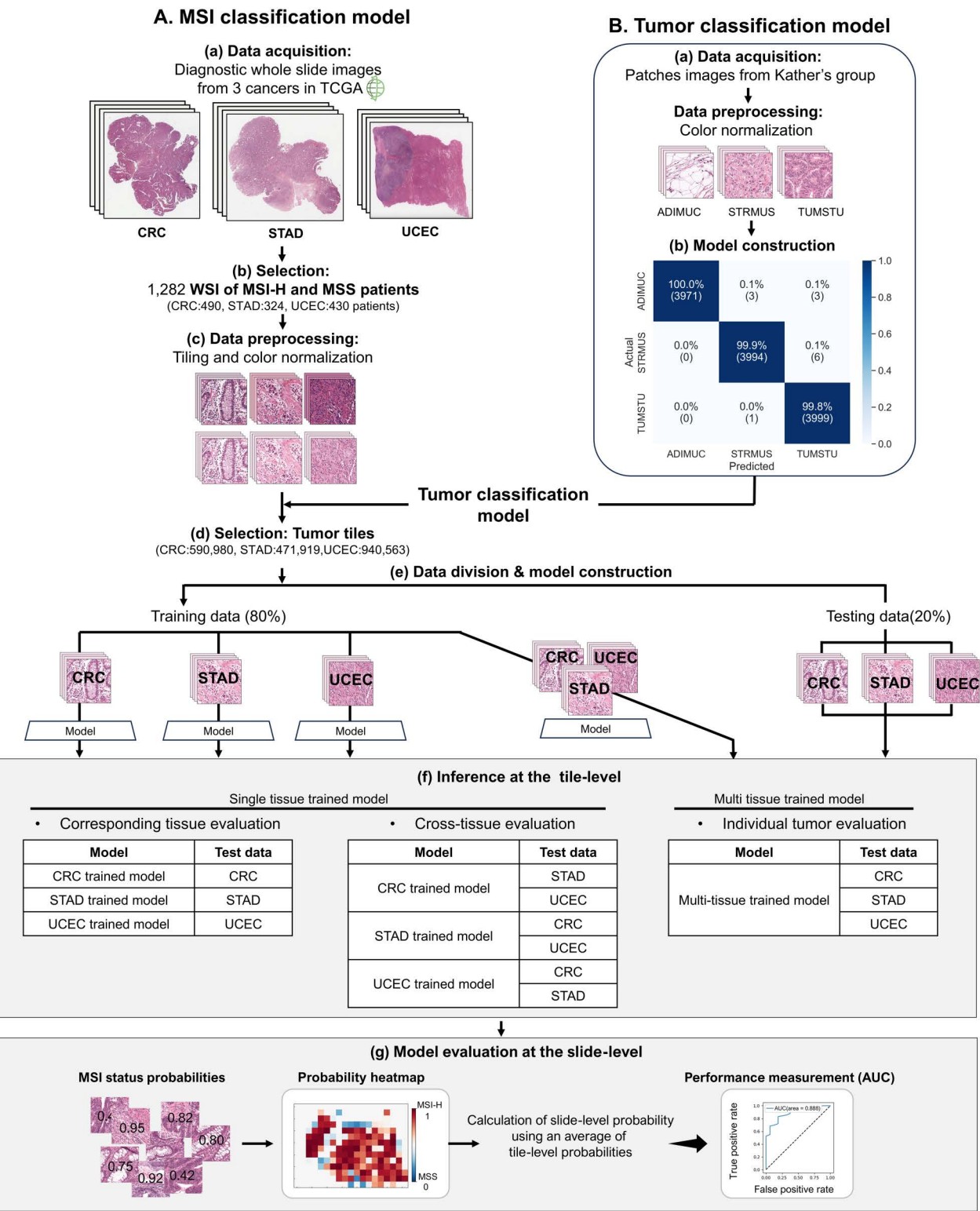

**Fig 1. Overall workflow. A. MSI classificaiton model development.** (a) H&E Whole slide image (WSI) of colorectal (CRC), stomach (STAD) and endometrial (UCEC) cancer were download from TCGA publich databases. (b) WSI of patients diagnosed with MSI-H and MSS using PCR testing were selected. (c) WSI were cut into 521×521 pixel tiles and color normalized by Macenko's method. (d) Tumor tiles were selected from the entire patches by

the tumor classification model. (e) The training data was used to train the convolutional neural networks with a 5-fold cross-validation, and the testing set was used to evaluate trained models each cancer types. The four models were trained using combinations of individual cancer types or tissues as training data. (f) The generated models (CRC-, STAD-, UCEC-, Multi-tissue trained model) were inferred the MSS status at the tile level for each cancer tissue. (g) Each model calculated slide-level probabilities by averaging tile-level probabilities, and model evaluations were compared using AUC. B. Tumor classification model development. (a) Tile image collected by Katther et al. [13] were download from the publicly available website (doi.org/10.5281/zenodo.2530788) to pretrain a tumor tissue classifier. Tiles were color normalized by Macenko's method. (b) The classifier has excellent performance of classifying tissue (overall accuracy = 99.67%) and detecting tumor tumor tiles (accuracy = 99.8%).

dataset contained 11,977 images tiles (ADIMUC = 3,977, STRMUS = 4,000 and TUMSTU = 4,000). All image tiles were 521 × 521 pixels at 0.5 µm/px. After color normalized by Macenko's methods, all tiles were divided into an 80:20 ratio for training and testing, and the training datasets were randomly divided into 5 folds for cross-validation of the data (S1 Fig in S1 File). We employed ResNet50 model architecture that trained from ImageNet. The pretrained model was fine-tuned for H&E images with last four layers being trainable. The model was trained with the following hyperparameters: a batch size of 32, learning rate of $10^{-4}$ for 50 epoches. The Adam optimizer and cross-entropy loss functions were used. Augmentation was applied with 25 degree of rotation, 50% probability of vertical/horizontal flipping. Each classifiers were trained for each fold, and the hyperparameters were determined when the highest average accuracy were identified. Once the parameters were confirmed, the model was trained on the complete training dataset, and the accuracy on testing datasets was used to assess overall model performance.

## A MSI tile classification model development

With the tumor dataset, another deep-learning model was constructed to distinguish tumor tiles between MSI-H or microsatellite stable (MSS) for specified tissue types. Tumor tiles that were specifically diagnosed with either MSI-H or MSS were used as a dataset for model training. To ensure robust model performance and prevent overfitting, we implemented a comprehensive cross-validation strategy. For the initial data split, we allocated 80% of the patients to the training set and 20% to the testing set, maintaining the proportion of MSI-H and MSS patient consistent across both sets for each cancer type (TCGA-CRC, TCGA-STAD, and TCGA-UCEC) (Table 1). Within the training set, we further divided the data into 5 patient-level folds for 5-fold cross-validation for hyperparameter tuning and model selection (S1B Fig in S1 File). For every fold, the same number of patches was randomly selected for each class (S3 Fig in S1 File). We created separate models

**Table 1. Datasets for constructing the MSI classification model.**

| The number of patients | | TCGA-CRC | TCGA-STAD | TCGA-UCEC |
|---|---|---|---|---|
| **Total** | | **490** | **324** | **430** |
| Train | MSI-H | 66 | 51 | 123 |
| (80%) | MSS | 326 | 208 | 221 |
| Test | MSI-H | 16 | 13 | 31 |
| (20%) | MSS | 82 | 52 | 55 |
| The number of patches | | TCGA-CRC | TCGA-STAD | TCGA-UCEC |
| Total | | 277,609 | 252511 | 352,194 |
| Train | MSI-H | 88,566 | 84,710 | 84,710 |
| | MSS | 88,566 | 84,710 | 84,710 |
| Test | MSI-H | 20,123 | 19,726 | 58,091 |
| | MSS | 80,354 | 63,365 | 124,683 |

for each cancer type as well as a combined model using data from all three cancer types. For the combined model, we maintained uniform distributions across cancer types to prevent any single cancer type from dominating the learning process. MSI classification model was optimized using the EfficientNet-b0, ResNet18, VGG19 model architecture, which are widely used and well-known for showing good performance, especially in H&E images [20,21]. We extended our evaluation by incorporating two recent high-performing vision architectures: ConvNeXT and NAT (Neighborhood Attention Transformer) [22,23]. All models were trained from ImageNet pretrained weights, and we modified the models in three ways depending on whether the partial layers are trainable or whether additional class layers are added. All models were trained from ImageNet pretrained weights, and we modified the models in three ways depending on whether partial layers are trainable or whether class layers are added. Trainable layers are re-trained for H&E images comprising 20% to 30% of the entire model's layers. And, the last linear layers were replaced by one or more new linear layers to accommodate the prediction of binary classification (S1 Table in S1 File). The models were trained with the following hyperparameters: a batch size of 256 and learning rate of $10^{-5}$ for 30 epoches and the same number patches of each class were fed into each batch. The loss, augmentation, and other conditions were the same as those used in the optimization of the tumor tile classification model.

As described above, we constructed a total of 60 models based on the training datasets and customized models. These include three single-tissue trained models, each focused on TCGA-CRC, TCGA-STAD, and TCGA-UCEC, and one model trained on the combined data of these three tissues. For additional evaluation, we constructed two more models trained on the combined data from two tissue types – TCGA-CRC and TCGA-STAD.

### Classification accuracy assessment in slide level

With each model, the MSI prediction score for every tumor tile was calculated as a probability value between 0 and 1. The slide-level MSI probability was calcualted as the average of the probabilities of the all the tumor patches in the WSI. The receiver operating characteristics (ROC) curve at the slide level were plotted. The same procedure was repeated for other tissue types, including TCGA-STAD and TCGA-UCEC. To assess the most universal model for classifying MSI status using the generated models, we employed three evaluation methods. Firstly, we evaluated the model's performance on the tissue for which it was trained. Next, to assess whether the model can distinguish MSI classification in different tissues, we tested it with datasets from other types of tissues beyond the training data. Lastly, we evaluated the model trained on three different tumors for each specific cancer tissue. This is to confirm whether the model trained on various cancer tissues can effectively classify the characteristics of MSI rather than the specific tissue features as the dataset size increases.

In the external validation assessment, evaluations were conducted on CPTAC-COAD and CPTAC-UCEC tissues using all available models, except the STAD-only trained model. This was due to the absence of STAD images in the CPTAC dataset.

### Statistical analyses

To demonstrate the performance of each classifier, receiver operating characteristic (ROC) curves and their Area Under the Curve (AUC)s are presneted for all the classifiers. AUC is defined as the area under the sensitivity-(1-specificity) curve.

## Results

### Image pre-processing and tumor tiles classification

From 1,282 WSIs (498 TCGA-CRC, 343 TCGA-STAD, and 441 TCGA-UCEC), 4,208,343 tile images (1,180,025 TCGA-CRC, 1,951,990 TCGA-STAD, and 4,208,343 TCGA-UCEC) were retrieved after WSI pre-processing, including tiling and normalization. The tumor classifier was performed with an overall accuracy of 99.67% (Fig 1, confusion matrix), and tumor tissue patches with a tumor probability higher than 0.95 were selected for the construction of the subsequent MSI

classifier model (S2A Fig in S1 File). It was confirmed that patches were selected with varying numbers of distributions for each slide (S2B Fig in S1 File). After combining the selected tiles classified as tumors, upon visual inspection, it was confirmed that the resulting image appropriately delineated tumor areas based on annotated slides (Fig 2).

A total of 2,003,462 (590,980 TCGA-CRC, 471,919 TCGA-STAD, and 940,563 TCGA-UCEC) tumor tiles were filtered from the tumor classification model. Then, we limited the number of patches for MSS, selecting them randomly to address the data imbalance issue when constructing the MSI classification deep learning model. As a result of this process, some MSS patches were excluded and finally a total of 882,314 tiles were selected (Table 1 and S3 Fig in S1 File).

In the CPTAC datasets, 47,275 tumor regions were identified out of 101,811 tiles from 64 CPTAC-COAD WSIs, while 43,318 tumor regions were classified out of 56,372 tiles from 74 CPTAC-UCEC WSIs.

## MSI-H classification model evaluation in slide level

In the corresponding tumor evaluation, the EfficientNetb0 models showed better performance than other models in identifying MSI mutations in histopathological images at the slide level, with mean AUC values of 0.88 in TCGA-CRC, 0.80 in TCGA-STAD, and 0.66 in TCGA-UCEC. VGG19 showed mean AUC values of 0.82 in TCGA-CRC, 0.75 in TCGA-STAD, and 0.64 in TCGA-UCEC, and ResNet18 showed mean AUC values of 0.85 in TCGA-CRC, 0.68 in TCGA-STAD, and 0.64 in TCGA-UCEC, and ConvNext showed mean AUC values of 0.82 in TCGA-CRC, 0.79 in TCGA-STAD, and 0.66 in TCGA-UCEC, and NAT showed mean AUC values of 0.88 in TCGA-CRC, 0.75 in TCGA-STAD, and 0.62 in TCGA-UCEC. EfficientNetb0 Model3 exhibited a highest performance of 0.93 for TCGA-CRC. EfficientNetb0 Model1 and ConvNext Model3 showed performance results of 0.84 for TCGA-STAD. EfficientNetb0 Model1 showed performance results of 0.84 and 0.69 for TCGA-STAD and TCGA-UCEC, respectively (Table 2). Models generally perform best on test data

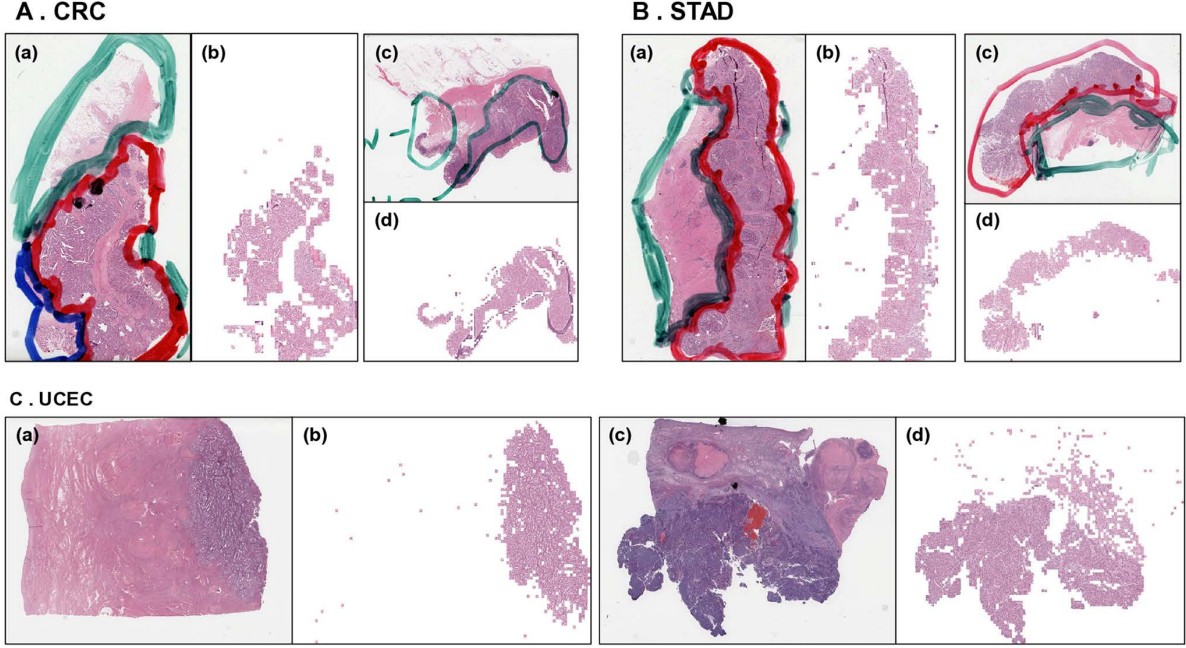

**Fig 2. Examples of whole slide image (WSI) and tumor tile predictions.** The WSIs were comprised both normal and tumor tissues **(a,c)**, which were divided into 521 × 521 pixel patches. Subsequently, utilizing a tumor classifier, only the tumor tissues were selected, and the resulting patches were integrated into a single image (b,d) on the side corresponding to the WSI for validation. Annotated regions of normal tissues are indicated by blue or green lines, while regions of tumor tissues are denoted by red lines (A-a and B-a,c). In the case of A-c, the green line represents the annotation region of the tumor.

**Table 2. The performance of our models.**

| Datasets | | EfficientNetb0 | | | ResNet18 | | | VGG19 | | | ConvNext | | | NAT | | |
|---|---|---|---|---|---|---|---|---|---|---|---|---|---|---|---|---|
| Train datasets | Test datasets | Model 1 | Model 2 | Model 3 | Model 1 | Model 2 | Model 3 | Model 1 | Model 2 | Model 3 | Model 1 | Model 2 | Model 3 | Model 1 | Model 2 | Model 3 |
| CRC | CRC | 0.89 | 0.82 | **0.93** | 0.87 | 0.83 | 0.86 | 0.79 | 0.76 | 0.9 | 0.81 | 0.78 | 0.87 | 0.89 | 0.87 | 0.88 |
| | STAD | 0.55 | 0.56 | 0.57 | 0.36 | 0.47 | 0.5 | 0.49 | 0.55 | 0.59 | 0.52 | 0.55 | 0.51 | 0.34 | 0.47 | 0.45 |
| | UCEC | 0.6 | 0.56 | 0.6 | 0.58 | 0.53 | 0.6 | 0.59 | 0.54 | 0.58 | 0.55 | 0.53 | 0.54 | 0.60 | 0.53 | 0.54 |
| | CPTAC COAD | 0.7 | 0.84 | 0.77 | 0.88 | 0.85 | 0.77 | 0.87 | 0.75 | 0.91 | 0.73 | 0.81 | 0.78 | 0.78 | 0.83 | 0.82 |
| | CPTAC UCEC | 0.46 | 0.48 | 0.5 | 0.5 | 0.48 | 0.54 | 0.43 | 0.32 | 0.33 | 0.42 | 0.42 | 0.46 | 0.54 | 0.39 | 0.5 |
| STAD | CRC | 0.72 | 0.79 | 0.81 | 0.71 | 0.75 | 0.7 | 0.69 | 0.59 | 0.66 | 0.76 | 0.79 | 0.74 | 0.79 | 0.77 | 0.79 |
| | STAD | **0.84** | 0.78 | 0.79 | 0.67 | 0.73 | 0.64 | 0.7 | 0.76 | 0.79 | 0.82 | 0.7 | **0.84** | 0.71 | 0.81 | 0.73 |
| | UCEC | 0.7 | 0.67 | 0.62 | 0.65 | 0.61 | 0.6 | 0.59 | 0.63 | 0.67 | 0.64 | 0.65 | 0.66 | 0.64 | 0.7 | 0.67 |
| UCEC | CRC | 0.57 | 0.51 | 0.53 | 0.6 | 0.6 | 0.58 | 0.6 | 0.62 | 0.57 | 0.61 | 0.61 | 0.61 | 0.63 | 0.67 | 0.62 |
| | STAD | 0.57 | 0.5 | 0.59 | 0.65 | 0.47 | 0.61 | 0.52 | 0.45 | 0.66 | 0.48 | 0.42 | 0.45 | 0.52 | 0.5 | 0.59 |
| | UCEC | **0.69** | 0.65 | 0.63 | 0.65 | 0.68 | 0.6 | 0.66 | 0.65 | 0.65 | 0.67 | 0.64 | 0.68 | 0.62 | 0.62 | 0.62 |
| | CPTAC COAD | 0.52 | 0.52 | 0.58 | 0.6 | 0.67 | 0.49 | 0.54 | 0.51 | 0.52 | 0.7 | 0.69 | 0.69 | 0.51 | 0.67 | 0.6 |
| | CPTAC UCEC | 0.72 | 0.68 | 0.72 | 0.72 | 0.68 | 0.76 | 0.6 | 0.55 | 0.63 | 0.64 | 0.6 | 0.63 | 0.69 | 0.56 | 0.75 |
| Multi-tissue | CRC | 0.83 | 0.8 | 0.8 | 0.83 | 0.8 | **0.87** | 0.7 | 0.7 | 0.77 | 0.81 | 0.83 | 0.78 | 0.86 | **0.87** | 0.83 |
| | STAD | 0.76 | 0.74 | 0.69 | 0.68 | 0.7 | 0.69 | 0.71 | 0.65 | 0.66 | 0.75 | 0.65 | 0.65 | **0.77** | 0.64 | 0.75 |
| | UCEC | 0.78 | 0.74 | 0.7 | 0.61 | 0.78 | 0.75 | 0.67 | 0.67 | 0.74 | 0.72 | 0.72 | 0.75 | 0.77 | **0.79** | 0.76 |
| | CPTAC COAD | 0.77 | 0.78 | 0.64 | 0.82 | 0.81 | 0.69 | 0.81 | 0.74 | 0.79 | 0.71 | 0.81 | 0.74 | 0.73 | **0.84** | 0.68 |
| | CPTAC UCEC | 0.61 | 0.66 | **0.77** | 0.53 | 0.65 | 0.73 | 0.43 | 0.44 | 0.65 | 0.65 | 0.49 | 0.61 | 0.66 | 0.59 | 0.63 |

This table presents the AUC values, with values in bold indicating the highest predictive performance in the respective cancer tissues.

that matches the tissue type on which they were trained. Performance tends to be lower when models trained on one tissue type were tested on a different type. EfficientNetb0 Model3 trained with TCGA-COAD tissue datasets obtained AUC values of 0.57 and 0.60 for TCGA-STAD and TCGA-UCEC in their respective test datasets. For the EfficientNetb0 Model1 trained with the TCGA-STAD dataset, the AUC value was 0.72 for TCGA-COAD and 0.57 for TCGA-UCEC test datasets. And the EfficientNetb0 Model1 trained with TCGA-UCEC datatsets predicted value was 0.57 for TCGA-COAD and 0.57 for TCGA-STAD test dataset (Table 2 and S4 Fig in S1 File).

To construct a multi-tissue trained model, we trained the models using the combination of TCGA-CRC, TCGA-STAD and TCGA-UCEC datasets. For TCGA-CRC, ResNet18 Model3 and NAT model2 showed the highest performance with an AUC of 0.87, while the VGG19 models generally showed lower performance results compared to other models. In TCGA-STAD, NAT and EfficientNetb0 Model1 have the good performance with an AUC of 0.77 and 0.76, respectively. These models for TCGA-CRC and TCGA-STAD showed lower or similar performances compared to models specifically trained for each tissue. However, the outcome of models for TCGA-UCEC showed a different patterns compared other tissues.. NAT Model2 exhibits the best performance for TCGA-UCEC with an AUC of 0.79, which is higher than the highest AUC of 0.69 achieved by a model trained on that corresponding tissue type alone (Table 2 and S5 Fig). Detailed model performance metrics, including accuracy, precision, recall, specificity, and F1 scores for all evaluated models, can be found in S2 Table in S1 File.

Analysis of the CPTAC validation dataset revealed that all models performed similarly on the CPTAC-COAD dataset, with results comparable to those obtained when trained on TCGA-CRC and tested on CRC. ResNet18 Model 1 achieved an AUC of 0.88, and VGG19 Model 3 reached 0.91, representing the highest performance. For the UCEC validation dataset, results were consistent with those observed when models were trained on TCGA-UCEC and tested on UCEC. In

multi-tissue testing, while all models demonstrated stable and consistent performance on COAD data, their performance was somewhat lower on UCEC data.

## Geographic visualization and comparative analysis of MSI prediction scores

We plotted heatmap and visualized tiles' MSI prediction value with their geographic location in WSI to understand whether the model effectively detects features and regions of MSI presentation. The MSI scores of slides were predicted by four models, TCGA-CRC-trained EfficientNet b0 Model3, TCGA-STAD-trained EfficientNetb0 Model1, TCGA-UCEC-trained EfficientNetb0 Model1, and multi-tissue trained EfficientNetbo Model1. Slides that matched the ground truth for MSI-H or MSS for each cancer type were selected. The models trained on the corresponding tissue and the multi-tissue trained model accurately predicted the distribution of MSI status areas, significantly consistent with the ground truth. In cross-tissue evaluation, the TCGA-CRC-trained model, for instance, exhibited values that diverged from the actual tissue output (0.64 MSI score for MSS in TCGA-STAD) or showed distant probability values (0.29 MSI score for MSI-H in TCGA-UCEC) in comparison to the ground truth labels (Fig 3). To understand which features the model ranks highest when distinguishing MSI-H from MSS, we pathologically analyzed regions with high probability values and regions with low probability values in the prediction heatmap. The results revealed that areas highly scored as MSI-H predominantly exhibited features of poorly differentiated carcinoma and high amounts of tumor infiltrating lymphocytes. This aligns with the histological characteristics of MSI-H known from existing pathological research [12,24]. Conversely, tissue regions highly scored as MSS displayed features of well to moderately differentiated carcinoma (Fig 4).

## Discussion

### MSI-H classification model evaluation

Our models achieved the hightest AUC values of 0.93, 0.84, and 0.79 in TCGA-CRC, TCGA-STAD and TCGA-UCEC among various our models, respectively. Previous studies utilized the TCGA dataset to develop and evaluate their DL models for MSI status through intra-study cross-validation. The Kather et al [13] reported AUC values for various cancer types, including TCGA-CRC (AUC = 0.77, 95% CI: 0.62–0.87), TCGA-STAD (AUC = 0.81, 95% CI: 0.69–0.90), and TCGA-UCEC (AUC = 0.75, 95% CI: 0.63–0.83). Similarly, the Bilal et al [25] demonstrated AUC of 0.86 ± 0.03 for TCGA-CRC, while the Guo et al [26] reported a high AUC of 0.91 ± 0.02 for TCGA-CRC. Comparing these results to our models trained on individual tissues or multi-tissues from TCGA-CRC, TCGA-STAD, and TCGA-UCEC, we observed that our models performance are either comparable or higher.

A model trained on corresponding tumor tissue showed higher accuracy compared to trained on other tissue types, indicating that tissue-specific features learned during training do not always generalize well to other tissue types. In the combined analysis, the multi-tissue trained model had lower AUC values for TCGA-CRC and TCGA-STAD compared to single-tissue trained analysis for TCGA-CRC and TCGA-STAD. However, for TCGA-UCEC, the performance actually increased when trained with multi-tissue. We anticipated that the muti-tissue trained models would generalize molecular morphologies distinguishing between MSS and MSI, leading increase performance in individual tissues. However, it did not imporove the performance in individual. We observed that models trained on TCGA-UCEC significantly underperformed compared to those trained on other tissues. We thitough that this might have a substantial impact on the overall performance of multi-tissue trained models. We constructed another multi-tissue trained model (TCGA-CRC+STAD) excluding TCGA-UCEC images and evaluated for each tissue type, seperately. In the muti-tissue trained EfficienNetb0 Model1, the achieved performances were an AUC of 0.83 in TCGA-CRC, 0.76 in TCGA-STAD, and 0.78 in TCGA-UCEC, while in the TCGA-CRC+STAD trained EfficientNetb0 Model1, they were 0.86, 0.76, and 0.75, respectively, showing similar levels of performance across different conditions (S6 Fig in S1 File). This observation underscores the consistent of the model's performance in learning diverse datasets.

Previous studies [20] showed that a combined model of TCGA-CRC+STAD (AUC 0.77) did not enhance performance over a model trained on TCGA-CRC (AUC 0.80) in detecting MSI in TCGA-CRC. This finding shows results similar to

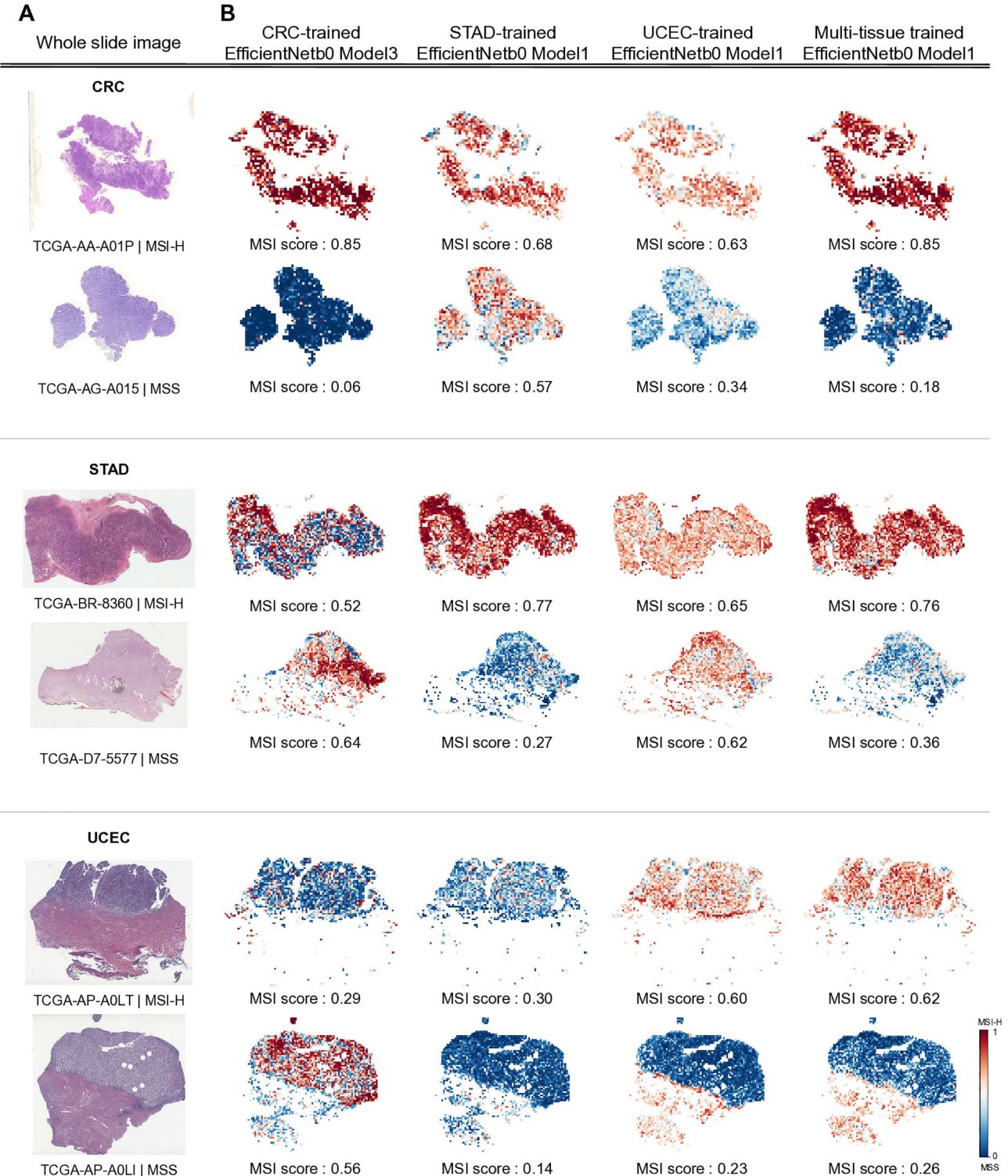

**Fig 3. Visualization of MSI probability heatmap at the slide level. A.** Whole slide images. **B.** Corresponding predicted MSI heatmaps for the image shown in A visualize patch-level MSI scores generated by three single-tissue trained models and a CRC-STAD-UCEC tissue trained model. The average patch-level MSI score beneath each heatmap represents the slide's MSI value. The heatmap bar illustrates MSI scores ranging from 0 to 1, where values closer to 1 indicate MSI-H and values closer to 0 suggest a higher probability of MSS.

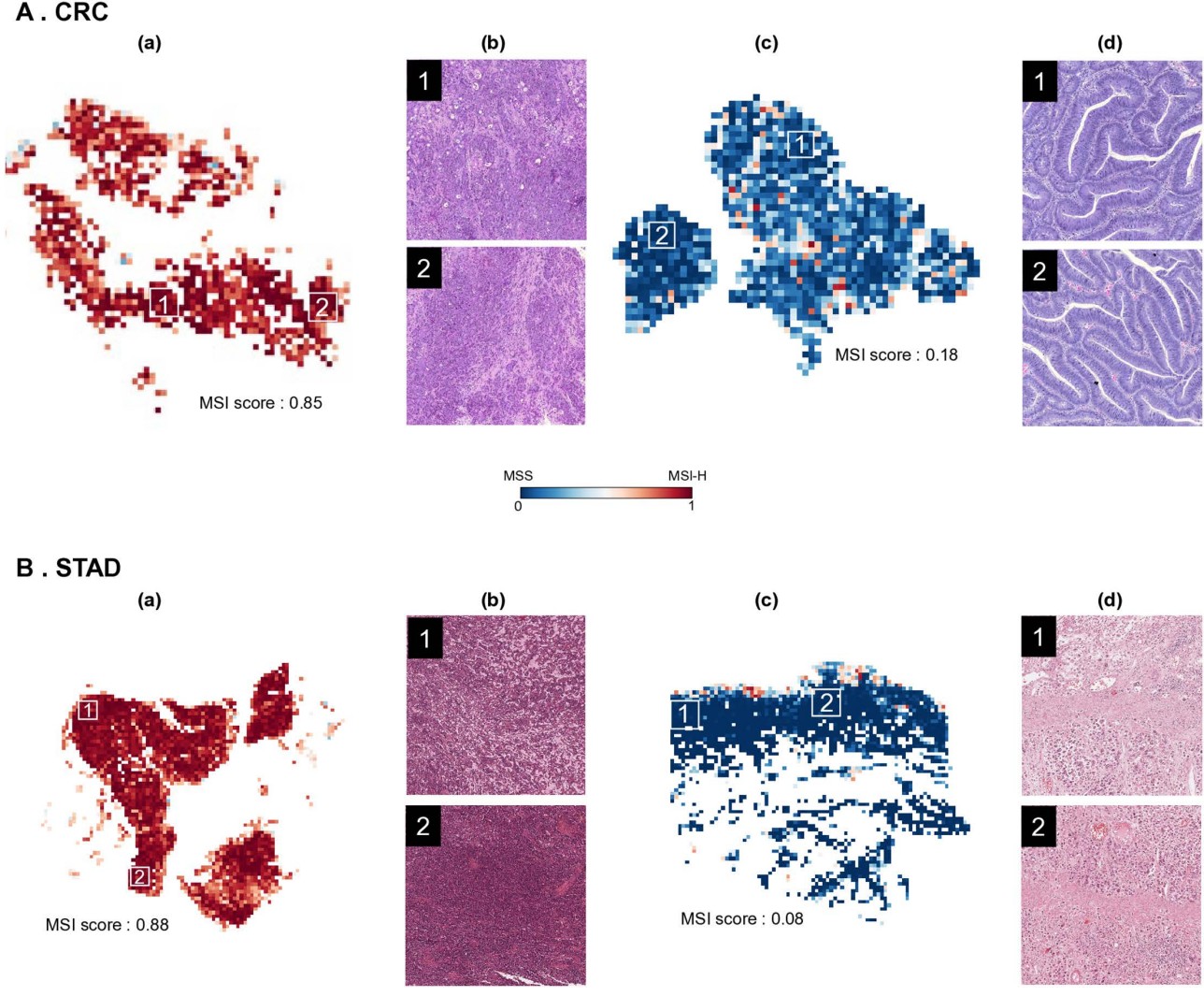

**Fig 4. Comparative examples of MSI prediction patterns and tissue morphology.** The prediction heatmaps (a and c) display results generated using an EfficientNet Model1 architecture with multi-tissue training. These maps show predicted microsatellite instability status across tissue samples, with corresponding H&E histology images (b and d) revealing the actual tissue morphology from regions marked by white boxes. A and B represent colorectal cancer and stomach cancer, respectively, with results showing microsatellite instability high (a, MSI scores: 0.85 and 0.88) and microsatellite stable (c, MSI scores: 0.18 and 0.08) status.

ours. They explained that there was because of different trends in debris, lymphocytes, and necrosis across each tissue image. Additionally, genomic analyses of TCGA-CRC and TCGA-UCEC revealed tissue-specific differences in the frequency of frameshift and in-frame MSI mutations among genes [27]. A recent models that distinguishes immune morphologies, trained on a combination of ten cancer tissues, reported enhanced performance (mean AUC 0.51–0.95) over individually trained models (mean AUC 0.59–0.77), suggesting that immune morphologies can be generalized at a multi-tissue level [28].

Models, that have effectively learned tissue-specific features and shown high performance in corresponding tissues, may show a decrease in performance when trained with additional images of different tissues, as this could neutralize the specific histological MSI features. Conversely, models, that were trained on their own data and exhibited low performance,

are presumed to have not adequately detected the general characteristics of MSI, including its specific tissue features. Adding images from other tissues for training may assist in detecting the general features of MSI, potentially leading to improved performance. When considering the performance differences between models that have precisely learned tissue-specific features and those trained on multi-tissue data, finding a balance between the model's generality and specialized precision remains a significant challenge for research.

Colorectal cancer, gastric cancer, and endometrial cancer each exhibit distinct molecular characteristics [29–32] and tumor microenvironments (TME) [33]. According to Tumor-Infiltrating Lymphocyte (TIL) mapping studies based on H&E images from TCGA samples, gastric cancer shows the highest TIL ratio at approximately 14.6% and exhibits histological features of immune response across more extensive regions compared to other cancer types [34]. Analysis of matrisome gene expression patterns, a key component of TME, across these three cancers reveals that colorectal and gastric cancers share similar expression patterns forming a single cluster, while endometrial cancer shows unique matrisome transcription factor regulation patterns distinct from the other cancer types [35]. Furthermore, TCGA cohort samples display diverse clinical characteristics, and tumor histological characteristics may vary due to biological differences among the medical centers where patients received treatment [36].

MSI detection across multiple cancer types presents several challenges. While MSI classification in a single cancer type involves binary classification between MSI and non-MSI within that cancer type, multi-tissue trained model classification requires the model to understand and learn diverse manifestations of MSI across different cancer types, resulting in a more complex decision-making process. These models face technical difficulties in distinguishing cancer-type differences and MSI status. Additionally, MSI-related features often appear as weak signals overshadowed by the dominant characteristics of each cancer type, making it challenging to identify subtle MSI patterns. To address these challenges, we evaluated the performance of various model architectures with different characteristics. However, the performance of multi-cancer model did not surpass that of single-cancer models, likely due to the complexities of multi-domain learning and increased task difficulty. To overcome these limitations, we suggest that future work should focus on developing specialized architectures that employ domain adaptation techniques (such as Domain Adversarial Neural Networks – DANN) to normalize cancer-type-specific histological features and align MSI feature distributions across cancer types [37].

In our study, we employed three distinct CNN architectures. VGG19 features a uniform structure with nineteen layers and 3×3 convolution filters, demonstrating exceptional capability in extracting detailed visual features. While its hierarchical feature learning excels at capturing subtle tissue patterns, the model faces challenges with gradient vanishing due to its deep structure and high computational costs stemming from its 140 million parameters [38]. ResNet18 addresses deep structure's gradient vanishing problem by introducing shortcut connections. Despite its relatively shallow 18-layer structure, its residual learning approach enables efficient learning of complex tissue patterns while reliably preserving important feature information [39]. EfficientNetb0 achieves high accuracy and efficiency through compound scaling methods that automatically adjust network width, depth, and resolution, combined with Neural Architecture Search. While particularly adept at processing various scales of features in histopathological images, it presents implementation challenges due to its complex architecture and risks overfitting on smaller datasets [40]. To leverage the advantages of recent attention mechanisms, we evaluated two state-of-the-art models. ConvNeXT incorporates transformer design principles into CNN architecture, utilizing 7×7 kernels instead of traditional 3×3 kernels, expanding channel capacity, and introducing Layer Normalization and Depthwise Convolution. However, this model risks overfitting on small datasets and may exhibit unstable transfer learning performance across different domains [22]. NAT effectively combines attention mechanisms with hierarchical processing, achieving a balance between local context preservation and computational efficiency while successfully implementing CNN strengths such as locality, translation equivariance, and hierarchical feature representation. However, its complex structure makes model tuning and optimization challenging for specific tasks, and performance can vary significantly depending on task characteristics [23]. This comprehensive analysis revealed distinct performance patterns across models, varying significantly with cancer type characteristics and transfer learning strategies.

In the performance analysis of multi-tissue training, distinctive patterns emerged across different validation datasets. EfficientNetb0 Model 1 achieved the highest and most stable performance on internal TCGA datasets, with a mean accuracy of 0.78 (SD = 0.04). In a broader evaluation including both internal TCGA and external CPTAC datasets, NAT Model 1 demonstrated the highest overall performance, achieving a mean accuracy of 0.76 (SD = 0.072), while EfficientNetb0 Model 2 exhibited the most robust generalization capability, with a mean accuracy of 0.74 (SD = 0.05).

These performance variations reflect the fundamental architectural differences of each model. The compound scaling methodology central to EfficientNetb0 effectively adjusts network depth, width, and resolution, enabling the capture of multi-scale features in medical images. This adaptability likely contributed to its stable performance across diverse tissue types. Meanwhile, the neighbor attention mechanism in NAT's architecture demonstrated a strong capability in integrating local features with global contextual information. This architectural advantage allowed for consistent extraction of crucial visual features from pathological images, even those collected from institutions with varying characteristics, contributing to NAT's strong performance across diverse datasets.

### Limitations and future work

Limitations exist in building models in this study. Among our models, EfficientNet models, partially trained with parameters suited for H&E images, exhibited superior performance. Nonetheless, our model exhibits both false positive and false negative results, and to enhance performance for individual tumors, better model construction is needed (Fig 5). In

**Fig 5. False results of microsatellite instability (MSI) prediction. A**. MSS falsely classifed as MSI-H (false positve). **B.** MSI-H falsely classfied as MSS (False netative). Left image is a Whole Slide Image, and the two images on the right are visualizations of MSI probability heatmaps at the slide level. The average patch-level MSI score beneath each heatmap represents the slide's MSI value. The heatmap bar illustrates MSI scores ranging from 0 to 1, where values closer to 1 indicate MSI-H and values closer to 0 suggest a higher probability of MSS.

future research, we aim to leverage foundation models pre-trained on H&E images instead of ImageNet, as they are more tailored to histopathological data. Additionally, we plan to explore multiple-instance learning (MIL) approaches to address weakly-supervised learning scenarios with limited label information more effectively.

The dataset utilized in this study does not include immunohistochemistry test results, another diagnostic method for MSI, making it currently impossible to compare these immunohistochemistry results with our deep learning model. Recognizing these limitations, in future research, we plan to establish an independent validation cohort to conduct direct comparisons between our deep learning model and various other diagnostic methods, including immunohistochemistry.

Additionally, UCEC model did not effectively predict the geographic region of MSI-H in the slide image, probably due to the training dataset of tumor tissue classification model, as the dataset only includes ground-truth of TCGA-COAD and TCGA-STAD tumor, but not TCGA-UCEC; therefore, it is uncertain that the tumor tiles classified in the TCGA-UCEC dataset are actual tumor tiles or tiles that just resemble features of colon and stomach cancer. Future studies can improve the TCGA-UCEC model by implementing a novel TCGA-UCEC tumor dataset or by manually labeling areas that show tumor areas.

## Conclusions

Our study attempted to construct an various models for MSI detection using datasets from multiple tissue types. Through the comparison of evaluations between models trained on multi-tissue and those trained on corresponding tissues, we observed diverse outcomes regarding which model demonstrated superior results depending on the type of tissue. There remains a challenge in finding a balance between the model's generality and specialized precision. However, our findings demonstrate the potential of multi-tissue trained models to identify features that can be generalized for MSI detection.

## Supporting information

**S1 File.    S1 Fig. Evaluation procedure and dataset division for tumor and MSI classifier models. S2 Fig. Tumor tissue probability and distribution per slide. S3 Fig. Datasets for train and test for the MSI classifier. S4 Fig. Comparing performances between the corresponding and cross tissue trained models. S5 Fig. Comparing performances between single-tissue and multi-tissue trained models. S6 Fig. Comparing performances between two-tissue and three-tissue trained models. S1 Table. Detailed model structure. S2 Table. Performance metrics.**
(ZIP)

## Author contributions

**Conceptualization:** Sejoon Lee, Jin-Haeng Chung.

**Data curation:** Jin-Ok Lee, Chang Yeon Kim.

**Funding acquisition:** Sejoon Lee, Jin-Haeng Chung.

**Software:** Jin-Ok Lee, Chang Yeon Kim.

**Supervision:** Sejoon Lee, Jin-Haeng Chung.

**Writing – original draft:** Jin-Ok Lee, Chang Yeon Kim.

**Writing – review & editing:** Sejoon Lee, Jin-Haeng Chung.

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
