## [Decision Letter · Decision Letter 0]

27 Sep 2024

Dear Dr. Lee,

Thank you for submitting your manuscript to PLOS ONE. After careful consideration, we feel that it has merit but does not fully meet PLOS ONE’s publication criteria as it currently stands. Therefore, we invite you to submit a revised version of the manuscript that addresses the points raised during the review process.

We look forward to receiving your revised manuscript.

Kind regards,

Eduardo Andrés-León

Academic Editor

PLOS ONE

Journal Requirements:

1. When submitting your revision, we need you to address these additional requirements. Please ensure that your manuscript meets PLOS ONE's style requirements, including those for file naming. The PLOS ONE style templates can be found at https://journals.plos.org/plosone/s/file?id=wjVg/PLOSOne_formatting_sample_main_body.pdf and https://journals.plos.org/plosone/s/file?id=ba62/PLOSOne_formatting_sample_title_authors_affiliations.pdf 2. We note that the grant information you provided in the ‘Funding Information’ and ‘Financial Disclosure’ sections do not match.  When you resubmit, please ensure that you provide the correct grant numbers for the awards you received for your study in the ‘Funding Information’ section. 3. Thank you for stating the following financial disclosure: "This work was supported by a National Research Foundation of Korea (NRF) grant funded by the Korean Government (MSIT) (grant no. NRF-2021R1C1C1013706), and a research fund from Seoul National University Bundang Hospital (grant no. 14-2018-0013)" Please state what role the funders took in the study.  If the funders had no role, please state: "The funders had no role in study design, data collection and analysis, decision to publish, or preparation of the manuscript." If this statement is not correct you must amend it as needed. Please include this amended Role of Funder statement in your cover letter; we will change the online submission form on your behalf.

Additional Editor Comments:

Both reviewers have evaluated the article and believe it could be published in our journal. However, they have some concerns and questions about specific parts of the article. Therefore, I ask you to carefully read the reviewers’ comments and respond to each of their concerns, as well as revise the sections they consider necessary

Reviewers' comments:

**Comments to the Author**

1. Is the manuscript technically sound, and do the data support the conclusions?

Reviewer #1: Yes

Reviewer #2: Yes

2. Has the statistical analysis been performed appropriately and rigorously?

Reviewer #1: Yes

Reviewer #2: Yes

3. Have the authors made all data underlying the findings in their manuscript fully available?

Reviewer #1: Yes

Reviewer #2: Yes

4. Is the manuscript presented in an intelligible fashion and written in standard English?

Reviewer #1: Yes

Reviewer #2: Yes

Reviewer #1: This is an important and relatively well performed study. However, there is no new innovation in this study and no outstanding technical or medical contribution. There have been many studies using TCGA dataset for MSI prediction on histologic images using CNN models. The models authors used were relatively outdated and do not contain any novelty. There is no external validation on any other public dataset such as CPTAC or PAIP. Authors would want to consider applying their models to other public datasets or other dataset from SNU. Multiple instance learning or transformer-based learning can be considered for novel technical approach.

Reviewer #2: Multi-cancer analysis of histopathologic MSI screening based on digital histology image

This paper focuses on developing and evaluating deep learning models to detect microsatellite instability (MSI) using whole-slide images (WSI) from the TCGA dataset. The study targets three cancer types: colorectal cancer (CRC), stomach adenocarcinoma (STAD), and uterine corpus endometrial carcinoma (UCEC), utilizing convolutional neural networks (CNNs) like EfficientNet, ResNet18, and VGG19 to differentiate between high microsatellite instability (MSI-H) and microsatellite stable (MSS) tumor tiles. The results show that models perform best when tested on tissue types that match their training data, while performance drops when models are tested on different tissue types. The EfficientNet models outperformed the others, and the study found that while multi-tissue trained models sometimes improved performance, they did not always outperform single-tissue models. The paper highlights the challenge of balancing model generality and specificity in MSI detection. Future work aims to incorporate more advanced deep learning models and validate them with external datasets. There are two questions for the author and after doing minor changes, the paper can be accepted by Plos One.

1. Expand the Discussion on CNN Architectures: The authors should provide a more in-depth discussion on the impact of different CNN architectures on model accuracy. Specifically, it would be beneficial to explain how the structural differences between EfficientNet, ResNet18, and VGG19 affect the model's ability to detect MSI. Including insights on how these architectures handle variations in histopathological features and why one may outperform the others in certain tissue types would enhance the technical understanding.

2. Add Experiments to Investigate Generalization Issues: The authors should conduct additional experiments to explore the underlying reasons for the model's limited generalization across tissue types. Investigating the role of tissue-specific features, differences in tumor microenvironments, or variations in data distribution could provide valuable insights. These experiments would help identify the factors that hinder the model's ability to generalize and offer potential solutions to improve its performance in cross-tissue predictions.

**Do you want your identity to be public for this peer review?** For information about this choice, including consent withdrawal, please see our Privacy Policy

Reviewer #1: No

Reviewer #2: No

---

## [Author Response · Author response to Decision Letter 1]

11 Dec 2024

Dear Editor and Reviewers,

We greatly appreciate your thorough review and insightful suggestions that have helped improve our manuscript. We have carefully considered all comments and suggestions provided by the reviewers, and our point-by-point responses to each reviewer's comments are included in the attached file. Additionally, we have made appropriate revisions to improve the overall quality of the manuscript.

Sincerely,

Seejoon Lee

---

## [Decision Letter · Decision Letter 1]

23 Jan 2025

Dear Dr. Lee,

Thank you for submitting your manuscript to PLOS ONE. After careful consideration, we feel that it has merit but does not fully meet PLOS ONE’s publication criteria as it currently stands. Therefore, we invite you to submit a revised version of the manuscript that addresses the points raised during the review process.

We look forward to receiving your revised manuscript.

Kind regards,

Eduardo Andrés-León

Academic Editor

PLOS ONE

Additional Editor Comments :

A reviewer has requested a major revision of the manuscript. Please address all their comments thoroughly and respond to each point individually.

Reviewers' comments:

Reviewer's Responses to Questions

**Comments to the Author**

Reviewer #2: All comments have been addressed

2. Is the manuscript technically sound, and do the data support the conclusions?

Reviewer #2: Yes

3. Has the statistical analysis been performed appropriately and rigorously?

Reviewer #2: Yes

4. Have the authors made all data underlying the findings in their manuscript fully available?

Reviewer #2: Yes

5. Is the manuscript presented in an intelligible fashion and written in standard English?

Reviewer #2: Yes

Reviewer #2: This research proposes a deep learning model for detecting microsatellite instability in cancer diagnoses using whole-slide images from different types of cancers, specifically colorectal, stomach, uterine corpus, and endometrial adenocarcinomas. Differentiating high MSI (MSI-H) cases from microsatellite stable (MSS) cases was the primary objective. Public dataset images were used in this study, which trained models specifically for each cancer type and evaluated them on different and corresponding tissue types, as well as created a multi-tissue model. Major findings were high accuracy in the models specific to the tissues, with the highest for colorectal cancer and slightly lower for stomach and uterine/endometrial cancer. Multi-tissue models performed differently, though they showed promise in terms of generalizability across the different cancers. Despite the potential of MSI as a therapeutic target, traditional diagnostic methods like PCR and immunohistochemistry are costly and time-consuming. The study suggests that deep learning, particularly through analysis of WSI, could offer a quicker, cost-effective alternative for MSI screening, potentially enhancing patient prognosis by facilitating earlier and more accurate diagnoses.

I have several questions for this article:

1. Which of these features does the model rank highest when distinguishing MSI-H from MSS? Is it possible to use interpretability tools such as LIME or SHAP to visualize and understand these features? I suggest the author writes more to extend the Section “Geographic visualization and comparative analysis of MSIprediction scores”.

2. I suggest that the author could Implement robust cross-validation techniques to ensure the models are not overfitting, such as k-fold cross-validation or stratified splits based on cancer types.

3. How do the diagnostic accuracies of deep learning models compare with those of traditional methods—such as PCR and immunohistochemistry—in terms of sensitivity, specificity, and overall diagnostic yield?

**Do you want your identity to be public for this peer review?** For information about this choice, including consent withdrawal, please see our Privacy Policy

Reviewer #2: No

---

## [Author Response · Author response to Decision Letter 2]

8 Mar 2025

Dear Reviewer

We thank you and the reviewers for your time and consideration regarding our manuscript, PONE-D-24-10966. In the point-by-point responses below, we have addressed each of the referees’ comments.

---

## [Decision Letter · Decision Letter 2]

21 Aug 2025

Dear Dr. Lee,

Thank you for submitting your manuscript to PLOS ONE. After careful consideration, we feel that it has merit but does not fully meet PLOS ONE’s publication criteria as it currently stands. Therefore, we invite you to submit a revised version of the manuscript that addresses the points raised during the review process.

We look forward to receiving your revised manuscript.

Kind regards,

PLOS ONE

Journal Requirements:

Reviewers' comments:

Reviewer's Responses to Questions

**Comments to the Author**

Reviewer #2: All comments have been addressed

Reviewer #3: All comments have been addressed

2. Is the manuscript technically sound, and do the data support the conclusions?

Reviewer #2: Yes

Reviewer #3: Yes

3. Has the statistical analysis been performed appropriately and rigorously?

Reviewer #2: Yes

Reviewer #3: Yes

4. Have the authors made all data underlying the findings in their manuscript fully available?

Reviewer #2: Yes

Reviewer #3: Yes

5. Is the manuscript presented in an intelligible fashion and written in standard English?

Reviewer #2: Yes

Reviewer #3: Yes

Reviewer #2: This is a well-structured and methodologically sound study addressing a clinically relevant problem: using deep learning to predict microsatellite instability (MSI) from standard histology slides across multiple cancer types. The paper is clearly written, the experiments are comprehensive, and the discussion is thoughtful and balanced. The authors systematically compare models trained on single cancer types, evaluate their cross-cancer generalizability, and test a combined multi-cancer model. The inclusion of an external validation cohort (CPTAC) significantly strengthens the findings. The conclusion that multi-tissue models can improve performance for some cancers (UCEC) while not for others (CRC, STAD) is a nuanced and important contribution to the field.

The manuscript is of high quality and suitable for publication, pending minor revisions to enhance clarity and address a few key points.

Throughout: The term "hypterparameters" is used several times (e.g., line 151, 154, 184); it should be "hyperparameters".

Line 90: "staomach" should be "stomach". (Also seen in Figure 1).

Line 101: "publicy" should be "publicly".

Line 149: "emplolyed" should be "employed".

Line 174: "arcituecture" should be "architecture".

Line 189: "cutomized models" should be "customized models".

Line 212: "demonstarate" should be "demonstrate". "calssifier" should be "classifier". "chracteriestic" should be "characteristic".

Line 221: "a overall accuracy" should be "an overall accuracy".

Line 237: "classifcation model" should be "classification model".

Line 269 (Table 2 Title): "performnace" should be "performance".

Line 351: "We thougth that is may have" could be rephrased for clarity, e.g., "We thought that this might have..."

Reviewer #3: Thanks for making these revisions. I am satisfied with the current version. The authors may consider including relevant citations on image-guided cancer research to strengthen the study's background, such as:"

*DOI: 10.1016/j.cpsurg.2025.101819*

*DOI: 10.1016/j.cpsurg.2025.101833*

*DOI: 10.1016/j.cpsurg.2025.101817*

*DOI: 10.1016/j.cpsurg.2024.101640*

**Do you want your identity to be public for this peer review?** For information about this choice, including consent withdrawal, please see our Privacy Policy

Reviewer #2: No

Reviewer #3: No

---

## [Author Response · Author response to Decision Letter 3]

24 Aug 2025

In the point-by-point responses below, we have addressed each of the referees’ comments. We have conducted a comprehensive review of our reference list and can confirm that no retracted articles have been cited in our manuscript. All references have been verified as current and appropriate for our study.

Reviewer: 2

Response: Thank you for pointing out the typographical errors. We have carefully reviewed the manuscript and corrected all identified typos.

Reviewer: 3

Response: We thank the reviewer for the suggestion to include additional citations. Following this recommendation, we have added relevant content on image-based deep learning approaches for cancer research in the introduction, including several of the references to strengthen the study's background.

---

## [Editor Report · Decision Letter 3]

26 Aug 2025

Multi-cancer analysis of histopathologic MSI screening based on digital histology image

PONE-D-24-10966R3

Dear Dr. Lee,

We’re pleased to inform you that your manuscript has been judged scientifically suitable for publication and will be formally accepted for publication once it meets all outstanding technical requirements.

Kind regards,

Hao Zhang

Academic Editor

PLOS ONE
---

## [Editor Report · Acceptance letter]

PONE-D-24-10966R3

PLOS ONE

Dear Dr. Lee,

I'm pleased to inform you that your manuscript has been deemed suitable for publication in PLOS ONE. Congratulations! Your manuscript is now being handed over to our production team.

Kind regards,

on behalf of

Dr. Hao Zhang

Academic Editor

PLOS ONE